# Changing perioperative prophylaxis during antibiotic therapy and iterative debridement for orthopedic infections?

Lydia Wuarin[1], Mohamed Abbas[2], Stephan Harbarth[2], Felix Waibel [3], Dominique Holy[4,5], Jan Burkhard[4,5], Ilker Uçkay [1,2,3,4,5,6] *

1 Orthopedic Surgery Service, Geneva University Hospitals, Geneva, Switzerland, 2 Infection Control Program, Geneva University Hospitals, Geneva, Switzerland, 3 Department of Orthopedic Surgery, Balgrist University Hospital, Zurich, Switzerland, 4 Internal Medicine, Balgrist University Hospital, Zurich, Switzerland, 5 Infectiology and Infection Control, Balgrist University Hospital, Zurich, Switzerland, 6 Unit for Clinical and Applied Research, Balgrist University Hospital, Zurich, Switzerland

* ilker.uckay@balgrist.ch

## Abstract

### Background

Perioperative antibiotic prophylaxis in non-infected orthopedic surgery is evident, in contrast to prophylaxis during surgery for infection. Epidemiological data are lacking for this particular situation.

### Methods and findings

It is a single-center cohort on iterative surgical site infections (SSIs) in infected orthopedic patients. We included 2480 first episodes of orthopedic infections (median age 56 years and 833 immune-suppressed): implant-related infections (n = 648), osteoarticular infections (1153), and 1327 soft tissue infections. The median number of debridement was 1 (range, 1–15 interventions). Overall, 1617 infections (65%) were debrided once compared to 862 cases that were operated multiple times (35%). Upon iterative intraoperative tissue sampling, we detected pathogens in 507 cases (507/862; 59%), of which 241 (242/507; 48%) corresponded to the initial species at the first debridement. We witnessed 265 new SSIs (11% of the cohort) that were resistant to current antibiotic therapy in 174 cases (7% of the cohort). In multivariate analysis, iterative surgical debridements that were performed under current antibiotic administration were associated with new SSIs (odds ratio 1.6, 95%CI 1.2–2.2); mostly occurring after the 2nd debridement. However, we failed to define an ideal hypothetic prophylaxis during antibiotic therapy to prevent further SSIs.

### Conclusions

Selection of new pathogens resistant to ongoing antibiotic therapy occurs frequently during iterative debridement in orthopedic infections, especially after the 2nd debridement. The new pathogens are however unpredictable. The prevention, if feasible, probably relies on

**Data Availability Statement:** All relevant data are within the manuscript and its Supporting Information files.

**Funding:** The authors received no specific funding for this work.

**Competing interests:** The authors have declared that no competing interests exist.

surgical performance and wise indications for re-debridement instead of new maximal pro-phylactic antibiotic coverage in addition to current therapeutic regimens.

## Introduction

The ideal regimen for perioperative antibiotic prophylaxis for prevention of surgical site infections (SSI) is evidence-based for the majority of clean, non-infected orthopedic procedures [1–4]. However, standard prophylaxis protocols do not recommend specific regimens before re-debridement of patients under already implemented curative antibiotic therapy for an established orthopedic infection (SSI or community-acquired) [1,2]. Scientific literature and epidemiological evaluations are lacking, but clinicians acknowledge that the microbiological spectrum may change during the course. The surgical debridement may itself cause new SSI; or a former SSI may get a new bacterial, postoperative SSI.

Practically, when performing a second look during ongoing antibiotic therapy, surgeons continue with the current therapeutic antibiotics or, if clinical evolution is unsatisfactory, empirically broaden the spectrum after obtaining new intraoperative tissue samples. Alternatively, few colleagues administer the standard perioperative prophylaxis, independently of the pathogens, simply because they lack specific protocols. New intraoperative cultures during re-operation may remain negative because of the influence of systemic antibiotics [5], but they might also grow previously unidentified pathogens typically resistant to current antibiotics.

These new pathogens indicate a dilemma. If clinical evolution is satisfactory, physicians might interpret them as a selection or contamination, and usually continue with the antibiotic treatment in place. However, besides a pre-planned re-intervention (in order to reduce the bacterial load surgically), mostly the evolution has been unsatisfactory; hence the indication for re-debridement. Consequently, these new pathogens are interpreted as new SSIs, with broadening of the spectrum and prolongation of total antimicrobial therapy [6].

In this study, we aimed to evaluate the missing epidemiology and specifically link the occurrence of new SSIs to the numbers of iterative re-debridement that we performed under current therapeutic antibiotic agents and. We wonder if these patients would profit from extended prophylaxis during re-debridement; and if the nature of possible secondary SSIs would be predictable.

## Methods

The Geneva University Hospitals is a tertiary center for septic orthopedic surgery and associated infectiology [7]. For the current study, we used a composite database 2004–2017 (Ethical Committee no. 13–178, 08–057 [8], 08–06 [9], and 14–198), including all adult patients hospitalized for clinically moderate and severe orthopedic infections, including the diabetic foot [10]). We did not collect tissue samples and did not contact the patients specifically for that study, but used their old anonymized data to compose our database.

We excluded cases that were amputated in toto [11], cases with antibiotic-free windows before re-debridement [5], and episodes for which the occurrence of newly identified pathogens did not change the antibiotic regimen, because we interpreted them as "contamination", because the newly detected bacteria had no clinical impact on the further management. In contrast, pathogens sensitive to original antibiotic therapy and presumably causative of clinical worsening, were identified as new pathogens. We defined infection as intraoperative pus and clinical signs of infection (color, calor, pain). SSI definitions based on the Center of Disease

Control standards [12]. We collected several microbiological samples from deep intraoperative tissues, and ignored results of superficial specimens or sinus tracts. We regrouped coagulase-negative staphylococci [13], micrococci, corynebacteria or propionibacteria as "skin commensals". We assessed the first five pathogens of semi-quantitative cultures and arbitrarily censored thereafter. The Microbiology Laboratory processed all specimens according to Clinical and Laboratory Standard's Institute recommendations [14], before switching to the EUCAST criteria (European Committee) in 2014 [15].

Of note, besides prior to the very first debridement for orthopedic infection (when the antibiotics were first started after intraoperative microbiological samplings), all study patients were under systemic antibiotic therapy. This therapy was either empirical or targeted to previously identified pathogens. In this manuscript, the term "prophylaxis" refers to a true perioperative antibiotic prophylaxis, which is only given as a single dose and is not continued after debridement; independent of current systemic antimicrobial therapy. In contrast, the clinical changing of antibiotic regimens after/during debridement would be a preemptive, or targeted, therapeutic change, continuing for several days or weeks.

### Statistical analyses

The primary objectives of this study were to determine possible mismatch between current curative antibiotic therapies and newly identified bacterial superinfection after debridement and to evaluate the need of a prophylactic antibiotic regimen, in addition to the ongoing curative antibiotic treatment. We performed group comparisons using the Pearson-$\chi^2$ or the Wilcoxon-ranksum-test. An unmatched multivariate logistic regression analysis determined associations with the outcome "SSI resistant to antibiotic therapy". We introduced independent variables in the univariate analysis stepwise into the multivariate analysis, except for the surgical and antibiotic-related parameters, which we forced into the final model. We computed the variables "total number of debridements", "number of debridements before new SSI", and the "time interval between consecutive debridement" as continuous and categorical variables. The cut-off values of the strata were chosen according to the middle stratum positioned around the median value of that variable. We further plotted new SSIs according to the number of prior debridements, and stratified new SSIs according to key pathogen groups. We used STATA software (9.0, STATA™, USA). $P$ values $\leq 0.05$ (two-tailed) were significant.

### Results

Overall, we included 2480 surgical patients with 2480 first episodes of adult orthopedic infections. The median age of the patients was 56 years (range, 18–99 y); 784 were females (32%) and 833 (34%) were immune-suppressed: diabetes mellitus (n = 454) [16], active cancer (113), severe alcoholism (68), medicamentous immune-depression (62), dialysis (25), cirrhosis CHILD C (17), solid organ transplantation (10), untreated HIV disease (5), agranulocytosis (4), splenectomy (1), pregnancy (1), or a combination of immune-suppressed states. We noted the following infections: implant-related infections (n = 648) [17]; osteoarticular infections (1153); 1327 soft tissue infections; and 213 diabetic foot infections [10]. We detected 83 different microbiological constellations during the initial assessment of infection and 273 newly acquired bacterial combinations on iterative surgeries. Overall, the five most frequently identified groups were *Staphylococcus aureus* (n = 1089; of which 148 methicillin-resistant *S. aureus*), streptococci (228), Gram-negatives (498; including 112 *Pseudomonas aeruginosa* cases [18], and skin commensals (304) [13]. The index pathogens were Gram-positive, Gram-negative [19], methicillin-resistant or skin commensals [19] in 1696 (68%), 498 (20%), 143 (6%), 453

(18%), and 304 (12%) cases, respectively. In 558 (22%) and 286 (12%) cases, initial assessments were polymicrobial and culture-negative [5].

## Iterative surgeries under curative antibiotic therapy

All patients were under systemic, curative antibiotic therapy for bacterial infection. We noted 867 different regimens prior to intraoperative samplings; divided upon administration route, changing during the course, combination therapies and different drug choices. An allocation of these 867 prior individual antibiotic regimens to the subsequent Overall, 1617 episodes (65%) were debrided once, compared to 862 cases with multiple debridements (35%); of which 510 a second time and 195 a third time. Formally, the median number of surgical debridement for infection was 1 (total range, 1–15 interventions; interquartile range, 1–2 interventions). The median delay between two consecutive interventions was 16 days. In 420 re-debridements (420/862; 49%), the current antimicrobial agent was continued without additional perioperative prophylaxis. In 90 cases, surgeons or anesthesiologists administered a supplementary standard prophylaxis with cefuroxime single dose 1.5 g intravenously [1–3,20] in addition to ongoing therapeutic antibiotics. Clinicians avoided to administer large-spectrum perioperative prophylaxis and avoided topical antibiotic prophylaxis regimens. Table 1 compares the study population with single vs. multiple debridements.

In this comparison, patients with bone and joint infections, implant infections, Gram-negative infections and infections due to skin commensals have been operated significantly more often than others, whereas sex, age, or immune-suppression did not influence the risk for re-operation.

## New pathogens and new susceptibility profiles according to the number of iterative surgeries

Among all iterative intraoperative samples during re-debridement, 507 were positive (507/862; 59%), but only 241 (242/507; 48%) returned a species already present in the index debridement. We witnessed thus 265 new pathogens (265/507; 52%) in the same patient. These new selections were Gram-positive in 192 cases and Gram-negative in 109 episodes and were interpreted as (new) SSIs, because of unsatisfactory evolution. As they were resistant to current antibiotics in 174 cases (174/507; 34%), clinicians broadened the therapeutic antimicrobial

**Table 1. Demographic and clinical variables comparing the second look to multiple debridements (>2 lavages).**

|  | Second look only |  | Multiple debridements |
|---|---|---|---|
| **n = 862** | **n = 509** | ***p* value*** | **n = 353** |
| Female sex | 142 (28%) | ***0.030*** | 123 (35%) |
| Age (median) | 58 years | 0.119 | 61 years |
| Immunosuppression[+] | 166 (33%) | 0.754 | 126 (36%) |
| Implant infections | 192 (38%) | ***0.013*** | 163 (46%) |
| Bone and joint infections | 305 (60%) | ***0.038*** | 236 (67%) |
| Diabetic foot infections | 37 (7%) | 0.100 | 16 (5%) |
| Polymicrobial infections | 108 (21%) | ***0.004*** | 105 (30%) |
| Initial Gram-positive infections | 367 (72%) | ***0.009*** | 225 (64%) |
| Initial Gram-negative infections | 97 (19%) | ***0.001*** | 105 (30%) |

* Significant *p* values ≤0.05 are displayed ***in bold and italic***.

[+] Immunosuppression = diabetes mellitus, corticosteroid medication, organ transplantation, cirrhosis CHILD C, dialysis, cancer, untreated HIV disease, alcohol dependency, pregnancy, agranulocytosis, splenectomy

spectrum and prolonged therapy. In contrast, 333 new pathogens were susceptible to the prior antibiotics. To cite an example, the overall proportion of methicillin-susceptible *S. aureus* among the causative pathogens had fallen from 38% to 11%, that of streptococci from 16% to 9% [21], while the proportion of methicillin-resistant *S. aureus* [8], enterococci [22], and non-fermenting rods [19] rose up significantly (Fig 1).

Stratified upon the groups of bone and joint infections, soft tissue and diabetic foot infections, the overall proportion of resistant new SSI were 13% (145/1153), 9% (120/1327), and 14% (30/213), respectively.

Table 2 shows clinical variables related to new antibiotic-resistant SSIs. The number of prior surgical debridements (all under current systemic antibiotic therapy) were significantly associated with the occurrence of new pathogens; independent of the initial pathogens. These new resistant SSIs were unpredictable regarding the microbiology and distributed among the entire Gram-positive and Gram-negative spectrum (Table 2; Fig 2A) with, however, a tendency towards more Gram-negatives with increasing numbers of surgical interventions, age, and a shorter delay between consecutive debridement (Table 3).

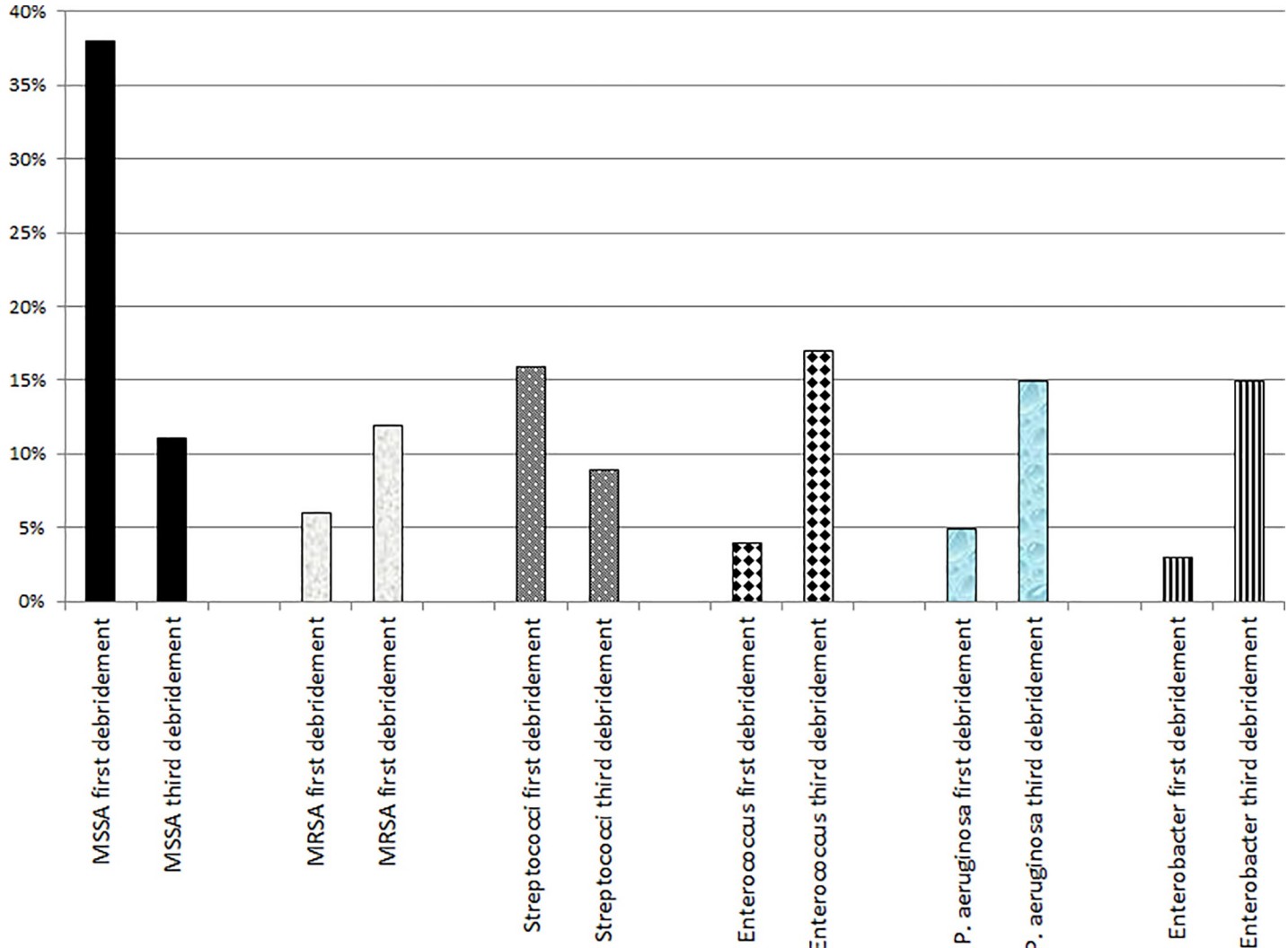

**Fig 1. Changes of intraoperative pathogens (selected examples) from the first debridement to the third debridement for the same orthopedic infection.**
MSSA = Methicillin-susceptible *Staphylococcus aureus*. MRSA = Methicillin-resistant *Staphylococcus aureus*. P. aeruginosa = *Pseudomonas aeruginosa*.

**Table 2. Characteristics of resistant pathogens in repetitive intraoperative samples performed under current antibiotic therapy** *(Some episodes have mixed new Gram-positive and Gram-negative infections, which we display separately in both lateral columns).*

| n = 283 | New Gram-positives n = 192 | *p* value* | Absence of new germs, n = 91 (Gram-positive) | Absence of new germs, n = 174 (Gram-negative) | *p* value* | New Gram-negatives n = 109 |
|---|---|---|---|---|---|---|
| Female sex | 57 (30%) | 0.606 | 701 (32%) | 701 (32%) | 0.763 | 36 (33%) |
| Age (median) | 63 years | 0.250 | 56 years | 56 years | 0.413 | 59 years |
| Immunosuppression+ | 73 (38%) | 0.146 | 732 (33%) | 732 (33%) | *0.042* | 50 (46%) |
| Median numbers of prior debridements | 2 | *0.035* | 1 | 1 | *0.028* | 2 |
| Prior (amino)penicillin therapy | 106 (55%) | 0.463 | 46 (51%) | 42 (24%) | 0.525 | 30 (28%) |
| Prior 1st-3rd generation cephalosporins | 21 (11%) | 0.789 | 9 (10%) | 20 (11%) | 0.537 | 10 (9%) |
| Prior glycopeptide & daptomycin therapy | 5 (3%) | 0.219 | 5 (5%) | 4 (3%) | 0.155 | 6 (6%) |
| Prior carbapenem & tazobactam therapy | 14 (7%) | 0.455 | 9 (10%) | 13 (7%) | 0.610 | 10 (9%) |
| Implant-associated infections | 65 (34%) | 0.740 | 29 (32%) | 65 (37%) | 0.062 | 29 (27%) |
| Osteoarticular infection | 106 (55%) | 0.463 | 46 (51%) | 100 (58%) | 0.109 | 52 (48%) |
| Initial polymicrobial infections | 86 (45%) | 0.760 | 39 (43%) | 73 (42%) | 0.343 | 52 (48%) |
| Initial Gram-positive infections | 109 (57%) | *0.001* | 1535 (69%) | 1535 (69%) | 0.128 | 68 (62%) |
| Initial Gram-negative infections | 66 (35%) | *0.001* | 413 (19%) | 413 (19%) | *0.001* | 37 (34%) |
| Initial skin commensal infections˚ | 32 (17%) | *0.034* | 256 (12%) | 256 (12%) | *0.032* | 20 (18%) |

* Significant *p* values ≤0.05 are displayed **in bold and italic**.

+ Immunosuppression = diabetes mellitus, corticosteroid medication, organ transplantation, cirrhosis CHILD C, dialysis, cancer, untreated HIV disease, alcohol dependency, pregnancy, agranulocytosis, splenectomy

˚ Skin commensals = coagulase-negative staphylococci, micrococci, corynebacteria or propionibacteria

Patients' sex, immune-suppression or localization of the orthopedic infections did not influence epidemiology. Table 4 summaries these new pathogens.

Many are naturally resistant to usual, narrow-spectrum β-lactam antibiotics (e.g. penicillins and 1st or 2nd generation cephalosporins). Of note, during the study period there was no specific outbreak in the septic orthopedic ward with the exception of five cases of vancomycin-resistant enterococci (VRE). The endemicity of methicillin-resistant *S. aureus* declined throughout the study period [8], and that of ESBL is rising [23]. Regarding the timing, new SSIs mostly peaked after the 2nd and 3rd debridement. Indeed, the microbiology during the first re-debridement still reveals two-thirds of known pathogens and one-third of new constellations. But already the second and third re-debridement switches to a third known pathogens and two-thirds of new ones (Fig 3).

## Multivariate adjustment

In view of the considerable case-mix, we adjusted with logistic regression analysis. We confirmed that with the occurrence of new antibiotic-resistant SSIs under current systemic antibiotic therapy and iterative surgeries (odds ratio 1.6, 95% confidence interval 1.2–2.2), (Table 5). Of note, since all patients undergoing iterative debridement were already under systemic antibiotic administration, we could not determine the impact of iterative surgeries alone (without concomitant antibiotic therapies) on the occurrence of these new SSI's.

Already the second debridement substantial under antibiotic treatment increased the odds ratio of new SSIs to twelve. In contrast, sex, age, and immune-suppression were unrelated. The

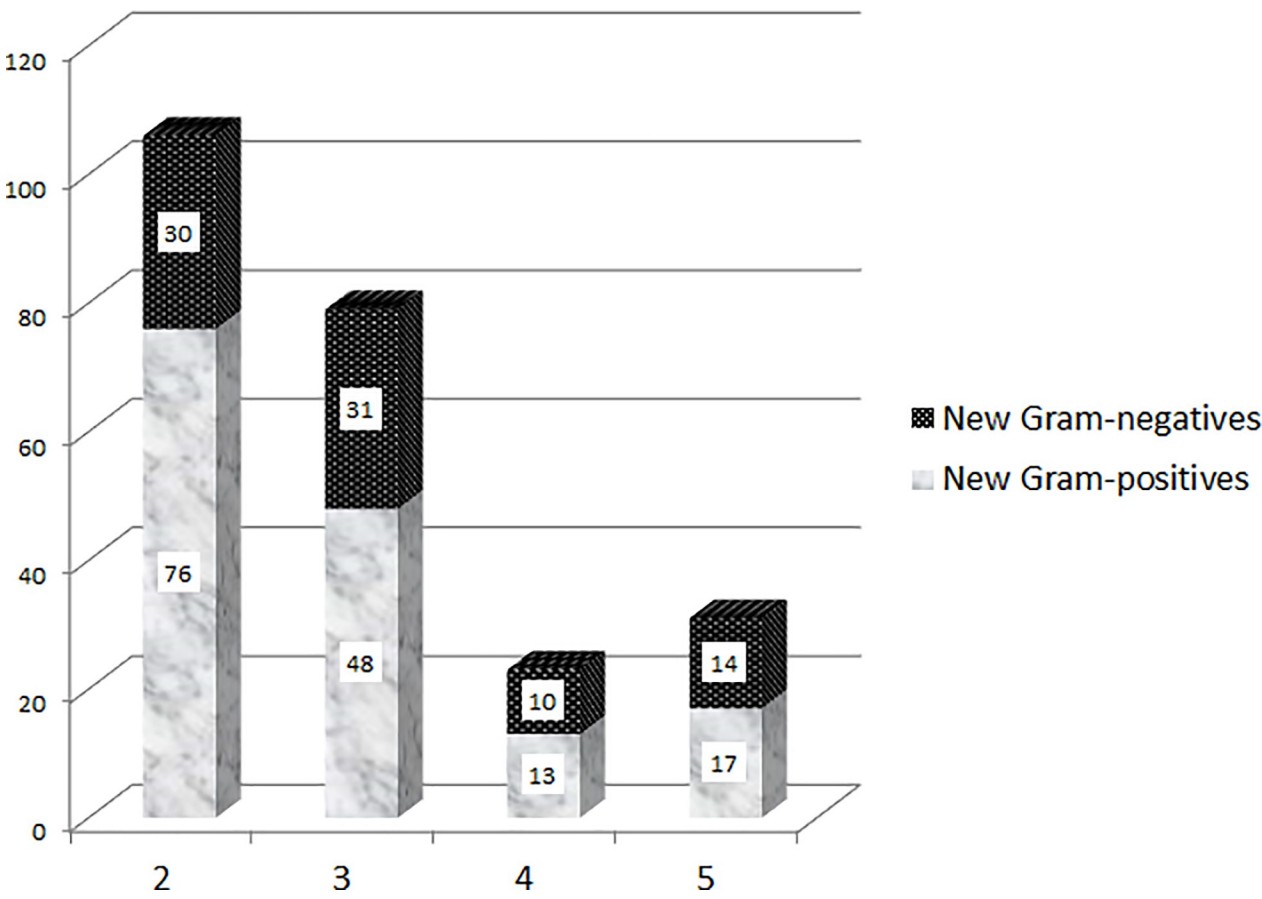

**Fig 2. Total number of new pathogens (vertical axis) stratified upon the Gram coloration and the number of debridement (horizontal axis).**

goodness-of-fit test was insignificant ($p = 0.41$) and the Receiver-Operating Curve value 0.86; showing a good accuracy of our final model.

**Table 3. Comparison between new Gram-positive and new Gram-negative surgical site infection under ongoing antibiotic therapy (Some episodes have mixed new Gram-positive and Gram-negative infections, which we display in both columns).**

|  | Gram-positive infections |  | Gram-negative infections |
| --- | --- | --- | --- |
| n = 301 | n = 191 | *p* value[*] | n = 109 |
| Female sex | 57 (30%) | 0.566 | 36 (33%) |
| Age (median) | 59 years | ***0.023*** | 61 years |
| Immunosuppression[+] | 73 (38%) | 0.195 | 50 (46%) |
| Implant infections | 65 (34%) | 0.182 | 29 (27%) |
| Bone and joint infections | 106 (55%) | 0.592 | 52 (48%) |
| Soft tissue infections | 85 (45%) | 0.592 | 57 (52%) |
| Median total number of debridements | 2 | ***0.001*** | 3 |
| Median number of debridements before new infection | 2 | ***0.001*** | 2 |
| Median delay between two debridements | 29 days | ***0.001*** | 22 days |

[*] Significant *p* values ≤0.05 are displayed ***in bold and italic***.

[+] Immunosuppression = diabetes mellitus, corticosteroid medication, organ transplantation, cirrhosis CHILD C, dialysis, cancer, untreated HIV disease, alcohol dependency, pregnancy, agranulocytosis, splenectomy

**Table 4. New pathogens and new orthopaedic surgical site infections during current antibiotic treatment (n = 273).**

| Gram-positives | Number | Gram-negatives | Number | Anaerobes and fungi | Number |
|---|---|---|---|---|---|
| Coagulase-negative staphylococci | 106 | Enterobacter | 31 | Bacteroides | 4 |
| Enterococci | 34 | Pseudomonas | 32 | Peptostreptococci | 4 |
| *Staphylococcus aureus* (susceptible) | 24 | *Escherichia coli* | 20 | Peptoniphilus | 1 |
| *Staphylococcus aureus* (resistant) | 22 | Klebsiella | 15 | | |
| Streptococci | 17 | Proteus | 14 | | |
| Corynebacterium | 9 | Morganella | 8 | | |
| Propionibacteria | 5 | Citrobacter | 5 | | |
| Bacillus | 3 | Serratia | 4 | | |
| Micrococci | 3 | Acinetobacter | 3 | | |
| Clostridium | 3 | Aeromonas | 2 | | |
| Actinomyces | 2 | Veillonella | 1 | | |
| | | Salmonella | 1 | | |
| | | Prevotella | 1 | | |
| | | Providencia | 1 | Candida | 5 |

## Discussion

This study provides insights in the complex epidemiology of iterative SSIs during multiple debridements and current antibiotic therapy for orthopedic infections. It is an original work, with a large number of patients included in an analysis from a retrospective database. Among 2480 adult patients, we re-debrided a third, and a quarter revealed new pathogens. Totally,

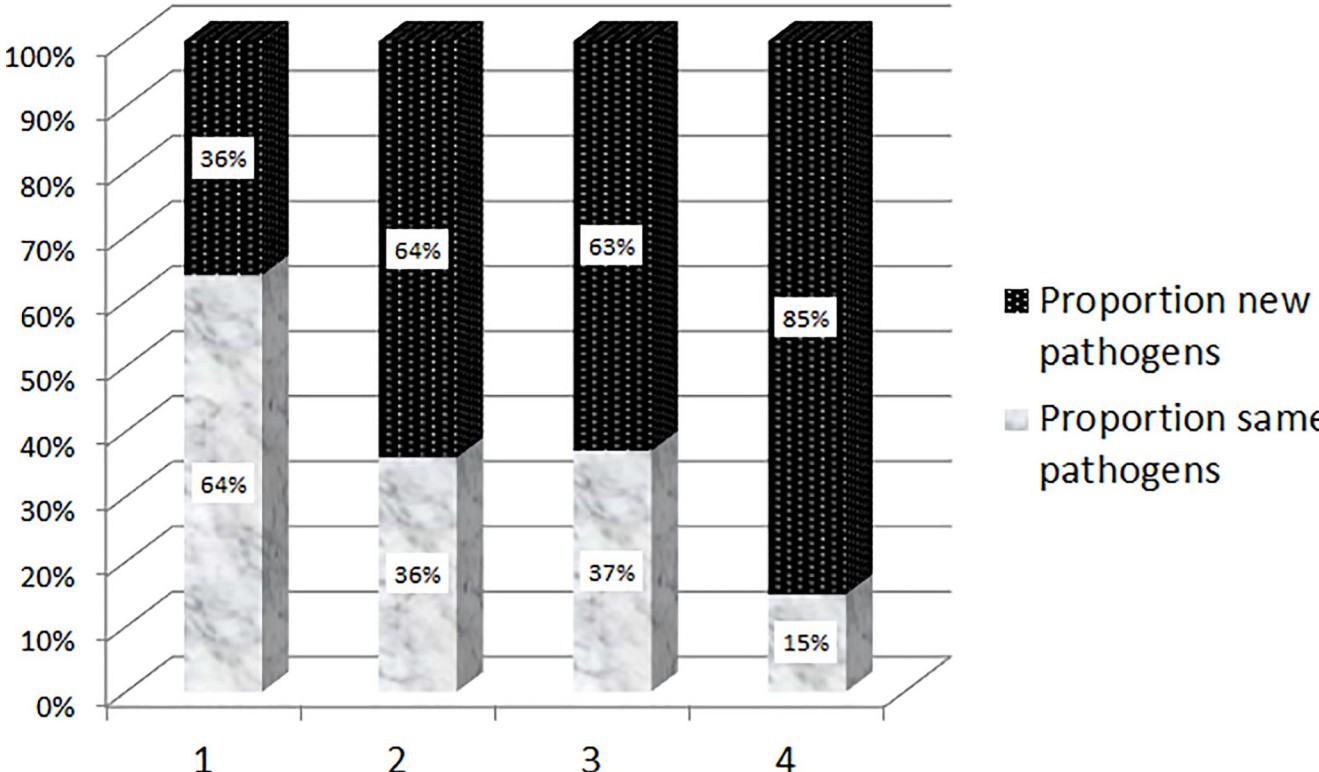

**Fig 3. Proportions of known versus newly identified pathogens (vertical axis) stratified upon the number of debridement (horizontal axis).**

**Table 5. Univariate and multivariate associations with resistant new SSI's** *(Logistic regression analysis; results expressed as odds ratios with 95% confidence intervals).*

| | Univariate analysis | Multivariate analysis |
|---|---|---|
| Female sex | 1.0, 0.7–1.3 | 0.9, 0.6–1.3 |
| Age | *1.0, 1.0–1.1*[*] | 1.0, 1.0–1.0 |
| Immunosuppression[+] | 1.2, 0.9–1.6 | 1.1, 0.8–1.6 |
| Implant infections | *1.6, 1.2–2.0*[*] | 0.9, 0.6–1.4 |
| Bone and joint infections | *1.4, 1.1–1.9*[*] | n.d. |
| Total number of debridements | *2.2, 2.0–2.4*[*] | *1.7, 1.3–2.1*[*] |
| - 2 debridements compared to 1 | *12.8, 8.3–19.8*[*] | *13.7, 8.8–21.2*[*] |
| - 3 debridements compared to 1 | *31.8, 19.8–52.1*[*] | *33.7, 20.7–54.6*[*] |
| - 4 debridements compared to 1 | *48.7, 30.0–79.4*[*] | *52.8, 31.9–87.4*[*] |
| No. of debridements until new infection | *1.2, 1.1–1.3*[*] | 0.9, 0.8–1.1 |
| - $\geq$ 1 debridements compared to 1 | *2.9, 2.0–4.1*[*] | n.d. |
| Time delay between two debridements | *1.0. 1.0–1.1*[*] | *1.0, 1.0–1.1*[*] |
| - 6–16 days compared to $\leq$ 5 days | *4.0, 2.3–7.5*[*] | *3.3, 1.7–6.3*[*] |
| - 17–46 days compared to $\leq$ 5 days | *8.1, 4.4–14.8*[*] | *5.6, 2.7–10.7*[*] |

[*] Statistically significant results are displayed *in bold and italic*. n. d. = not done

[+] Immunosuppression = diabetes mellitus, corticosteroid medication, organ transplantation, cirrhosis CHILD C, dialysis, cancer, untreated HIV disease, alcohol dependency, pregnancy, agranulocytosis, splenectomy

around ten percent of all episodes had new bacterial SSIs; with resistance to ongoing antibiotic agent in seven percent. From a clinical perspective, among 862 patients that required a re-debridement, 507 (59%) revealed a positive culture. In 265 (52%) the isolated microorganisms were different from the prior debridement. This means that from all episodes that required re-debridement, 30.7% (265) had a different pathogen. This is a major problem, particularly considered that the new microorganisms were often more resistant.

Since we only included relevant cases with immediate adaptation of the antibiotic therapy, we think that our interpretation of new SSIs is genuine and we are not facing mere selection and contamination. We think that it is nearly impossible to study our hypotheses in any other prospective and more controlled way. Moreover, the majority of the new microorganisms are undisputed pathogens for orthopedic SSIs [20] in Switzerland.

Available literature is very sparse. We identified only a single Spanish article with a similar study question, but in a very different setting. Ballus et al. published the epidemiology of surgical site peritonitis in an intensive care unit with broad-spectrum antibiotic use [6]. They prospectively observed 162 adult patients. Microorganisms isolated from tertiary peritonitis SSI's (160 cases; after combined surgical and medical treatment of previous episodes) revealed higher antibiotic-resistance (65%) than primary peritonitis. Every clinician would confirm this experience similar to our findings. Unfortunately, the authors lacked specific suggestions in terms of prevention of tertiary peritonitis, let alone concerning its optimal perioperative prophylaxis [6].

The legitime question is how much of these new SSI pathogens can be prevented by a modified or additional single-dose prophylaxis upon iterative debridement. The reason for a new SSI could be the consequence of miss-identification during the first surgery, new contamination during previous surgery for infection or superinfection of the wound on the ward despite current therapeutic antibiotic administration. Considering only the first two options as preventable, the third is not modifiable by any additional antibiotic administration. Clinically, the novel incidence of 7–11% SSIs warrants adaptation of perioperative prophylaxis for the first

and second conceptual situations. Standard second-generation cephalosporins or vancomycin [1–3] lack the necessary coverage in view of the random nature of the new pathogens. Unfortunately, we equally failed identifying a specific microbiological pattern to tailor a specific prophylaxis regimen. New postoperative superinfections appear Gram-positive, Gram-negative or both and include dozens of pathogen combinations; and this independently of initial pathogens, initial antimicrobial therapies, orthopedic infections or patient characteristics. An optimal total prophylactic coverage would hence theoretically consist of a combination of glycopeptides with aminoglycosides, or glycopeptides with carbapenems, piperacillin-tazobactam and similar spectra. Also, in some selected cases, a partial supplementary prophylaxis may be added on. For example, in patients treated with narrow-spectrum penicillin for streptococcal infections and multiple debridements, perhaps the combination with vancomycin might be sometimes indicated, but this is no maximal coverage by far and still needs to be proven as beneficiary.

However, unless there are future published clinical trials, we advocate against the introduction of this near-maximal prophylaxis because of the following reasons: First, perioperative prophylaxis is only one cornerstone of SSI prevention. It must be embedded in a whole bundle of measures [1–3]. Alone, it only reduces absolute SSI risks by some few percent [1]. Second, enhanced antibiotic prophylaxis lacks its final proofs, but might be associated with unnecessary adverse events (even when it is in single doses [24] or administered during three days such as in open fractures [4]). Several author groups proposed different enhancement strategies for non-infected orthopedic surgery: combining with local prophylaxis (e.g. local vancomycin in spine surgery [25]), double prophylaxis against Gram-negative [26] and Gram-positive [27] pathogens, or universal glycopeptide prophylaxis [28]. The majority of these enhancements failed to reduce SSI risk. Exceptions remain rare, very specific and often not reproducible by other research groups. At the same time, numerous reports documented transient kidney injuries by aminoglycosides [27] or combined vancomycin prophylaxis [28] in orthopedic surgery. Walker et al. reported that following a change in prophylaxis (from floxacillin & gentamycin to amoxicillin/clavulanic acid), they witnessed a 63% decrease in postoperative renal insufficiencies [29]. Moreover, enlarged prophylaxes, if implemented during a long period, could alter endemicity in septic orthopedics wards towards more resistant and Gram-negative pathogens [23].

Besides the fact that our study is retrospective, it has five major limitations. First, we ignore the acquisition route and the exact timing of the first presence and onset of the new SSI pathogens. We ignore if they were already colonizing the patient from the start, if they were present in the initial wound and subsequently selected by inactive antibiotics, or if they are true new acquisitions. Second, consequences of microbiological findings are arbitrary by nature. Infectious diseases physicians are often absent during surgery [7]. They have to decide the antibiotic changes, but are depending on the microbiological laboratory and especially upon the surgeons regarding clinical interpretation of the clinic and microbiological results (e.g. hematoma/seroma versus pus). Likewise, even if some new pathogens are clearly pathogenic, others might be not. Thus, in polymicrobial SSIs, it is quasi impossible to judge which of the pathogens is causative and which one is contamination. Moreover, new bacteria can also be a true new SSI that was simply not severe enough to worsen the clinical evolution. In that sense, when there is good clinical evolution, it is impossible to distinguish between colonization and clinical new infection. Third, although we analyzed many confounders, there are still some important variables unnoted, such as hand hygiene compliance [1], post-operative non-infectious wound complications [30] or use of negative-pressure vacuum therapy. Likewise, all patients undergoing iterative surgeries for infection, were already under systemic antibiotic therapy during re-debridement. Hence, we cannot pronounce on the possibility of

microbiological changes during iterative debridement in absence of antibiotic treatment. Fourth, in our study population, we had 83 different antibiotic regimens and an occurrence of 273 new pathogens. We moreover add a variety of 867 new antibiotic regimen changes throughout the therapeutic course in our study population. Such mixed constellations become too much detailed to be analyzed individually or to be individually displayed. We must recur to group analysis. Fifth, we limited the assessment of pathogens to the five most dominant ones in line with usual clinical practice. It is clear that a full microbiological work-up and a prolongation of the incubation time, beyond the standard five days, could alter overall epidemiological results.

## Conclusions

According to our cohort of 2480 adult patients with orthopedic infections, new SSIs occur at ten percent's risk during iterative debridement and concomitant antibiotic therapy. They already predominate after the 2nd debridement and are often resistant to administered antibiotics. Their microbial etiology seems unpredictable. We argue nevertheless against a total prophylactic coverage without prior prospective trials due to potential adverse effects and call for strict adherence to general infection control policies, evidence-based indications for surgical re-debridement and skilled surgical techniques [1–4]. The role of partial and selected enhancements of prophylaxis needs to be elucidated separately.

## Supporting information

**S1 File. Supporting Information files are uploaded.**
(XLSX)

## Acknowledgments

We thank the teams of the Laboratory of Bacteriology and the Orthopedic Service for support.

## Author Contributions

**Conceptualization:** Mohamed Abbas, Stephan Harbarth, Dominique Holy, Jan Burkhard, Ilker Uçkay.

**Data curation:** Lydia Wuarin, Mohamed Abbas, Ilker Uçkay.

**Formal analysis:** Lydia Wuarin, Ilker Uçkay.

**Investigation:** Stephan Harbarth, Ilker Uçkay.

**Methodology:** Mohamed Abbas, Stephan Harbarth, Felix Waibel, Ilker Uçkay.

**Project administration:** Lydia Wuarin, Ilker Uçkay.

**Resources:** Ilker Uçkay.

**Software:** Ilker Uçkay.

**Supervision:** Lydia Wuarin, Ilker Uçkay.

**Validation:** Stephan Harbarth, Ilker Uçkay.

**Visualization:** Felix Waibel, Jan Burkhard, Ilker Uçkay.

**Writing – original draft:** Lydia Wuarin, Felix Waibel, Dominique Holy, Jan Burkhard.

**Writing – review & editing:** Mohamed Abbas, Stephan Harbarth, Ilker Uçkay.

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
