## [Decision Letter · Decision Letter 0]

1 Nov 2019

PONE-D-19-23445

Changing perioperative prophylaxis during antibiotic therapy and iterative debridement for orthopedic infections?

PLOS ONE

Dear PD Dr Uçkay,

Thank you for submitting your manuscript to PLOS ONE. After careful consideration, we feel that it has merit but does not fully meet PLOS ONE’s publication criteria as it currently stands. Therefore, we invite you to submit a revised version of the manuscript that addresses the points raised during the review process.

We would appreciate receiving your revised manuscript by Dec 16 2019 11:59PM. To enhance the reproducibility of your results, we recommend that if applicable you deposit your laboratory protocols in protocols.io, where a protocol can be assigned its own identifier (DOI) such that it can be cited independently in the future. For instructions see: http://journals.plos.org/plosone/s/submission-guidelines#loc-laboratory-protocols

We look forward to receiving your revised manuscript.

Kind regards,

Daniel Pérez-Prieto, PhD

Academic Editor

PLOS ONE

Journal Requirements:

2. Please include your tables as part of your main manuscript and remove the individual files.

Please note that supplementary tables should be uploaded as separate "supporting information" files.

Reviewers' comments:

Reviewer's Responses to Questions

**Comments to the Author**

1. Is the manuscript technically sound, and do the data support the conclusions?

Reviewer #1: Yes

Reviewer #2: Yes

2. Has the statistical analysis been performed appropriately and rigorously? 

Reviewer #1: Yes

Reviewer #2: Yes

3. Have the authors made all data underlying the findings in their manuscript fully available?

Reviewer #1: Yes

Reviewer #2: Yes

4. Is the manuscript presented in an intelligible fashion and written in standard English?

Reviewer #1: Yes

Reviewer #2: Yes

5. Review Comments to the Author

Reviewer #1: The article titlted: “Changing perioperative prophylaxis during antibiotic therapy and iterative debridement for orthopedic infections?” Is a description of the microbiological findings in patients with orthopedic infections that required a new re-operation for failure of the infection during antibiotic treatment. This is a very original work, well written and with a large number of patients included in the analysis from a retrospective database. I have some comments:

1.- In the abstract and in your conclusions, you highlight that (abstract) “Selection of new pathogens resistant to ongoing antibiotic therapy occurs infrequently” (conclusion) “According to our cohort of 2480 adult patients with orthopedic infections, new SSIs occur at ten percent’s risk during iterative debridement and concomitant antibiotic therapy.” From a clinical perspective, I think that the impresive data is that from 862 patients that required a re-debridement 507 (59%) had a positive culture and in 265 (52%) the isolated microorganism is different from the first one. This means that from all that required a re-debridement 862, 30.7% (265) had a different pathogen and indeed this is not a minor problem particularly taken into account that the microorganisms were often more resistant.

2.- According to material and methods section: “We excluded cases that were amputated [11], cases with antibiotic-free windows before re- debridement…” so all patients included in the study were under antibiotic treatment. Then I dont understand what you say in results section: “Table 2 shows clinical variables related to new SSIs. Current antibiotic administration and…” All patients were under antibiotic treatment, so you mean a specific antibiotic? Indeed, in table 2 you show different antibiotics (penicillin, cephalosporins,…) but no one was significantly associated with new G+ or G-.

3.- In methods section for the multivariate analysis you mention: “An unmatched multivariate logistic regression analysis determined associations with the outcome “SSI resistant to antibiotic therapy”.” So until now the endpoint of the study (table 2) was new SSI and for the multivariate adjustment you change to antibiotic-resistant (to the current antibiotic treament) SSI. It is reasonable to expect that the majority of new pathogens were resistant to current antibiotics but this was indeed the case? Could you confirm that all new pathogens were resistant to current antibiotic treament. In addition, in multivariate adjustment you mention: “We confirmed that current antibiotic use was associated with new antibiotic-resistant SSIs (odds ratio 1.6, 95% confidence interval 1.2-2.2),…” but in table 5 there is no reference to current antibiotic variable??? Indeed, all the patients were under antibiotic treatment so I dont understand this as in point 2.

4.- The reason for a new infection (isolatation of a different pathogen from the first one) could be the consequence of 1.- miss identification during the first surgery for infection (unlikely considering that nowadays, and for sure in your expert hospital, several samples are collected during surgery, 2.- new contamination during previous surgery for infection (first or second or third… surgery) and 3.- superinfection of the wound after surgery. Considering only 2 and 3 as possible, the third is not modifiable by antibiotic prophylaxis but the number 2 could be significanlty reduced. In the text you talk about prophylaxis for the new debridement but this is not correct. New pathogens are indeed new infections so you have to talk about additional empirical treatment when a patient requires a new debridement, at least 30% had a new infection so in my opinion we have to recommend to broad the spectrum until deffinitive results are available (in general <3-5 days). A different point is whether the previous debridement to the current one with no pathogens, the same pathogen or a new pathogen, required a different prophylaxis to avoid contamination by a different pathogen (if the reason for a new infection is a contamiantion during the previous debridement, option number 2). Could you evaluate the antibiotics (prophylaxis, antibiotic treatment) received during the prior debridement to the one with or without a new pathogen? This would be really interesting.

Reviewer #2: Dear Authors,

I read with interest the paper. It is a comprehensive analysis of complex set of data of a large cohort of patients with orthopedic infection with a clear aim. It needs a huge effort to organize this type of data. And it is nearly impossible to study the study hypothesis in any other prospective and more controlled way. The aim of the study is clearly defined and important. The issue of antibiotic prophylaxis and treatment in unsuccessful orthopedic infections is a very important and still unsolved topic touching every specialist involved in treatment of this complex pathology.

Despite the complexity of the cohort data the authors were able to organize it in a systematic way. The presenting cohort is thus clearly outlined in the text and trough the tables.

Despite the study did not give a clear answer to the original question it still revealed the complexity of the clinical field and gave thought lines for the readers to integrate in their clinical decisions.

Some comments:

Page 3 last paragraph: Why were there revisions in satisfactory clinical evolution?

Page 4 last paragraph: Why episodes with new organisms that did not change the antibiotics were interpreted as contamination, just because they were sensitive to original antibiotic therapy? According to the definition of the infection it should thus be mentioned that only high grade infections were included.

Page 10 second paragraph: bundle of measures [1-3]. "Thus, avoiding unnecessary re-debridement, evidence-based surgical indications and techniques, and experienced surgical skills are certainly as important as mere addition of a new molecule." This sentence is very brave and to me lacking evidence besides common thinking.

Page 10 second paragraph:"At the same time, numerous reported transient kidney injuries by aminoglycosides [27] or combined vancomycin prophylaxis [28] in orthopedic surgery." The sentence has no verb.

6. PLOS authors have the option to publish the peer review history of their article (what does this mean?). If published, this will include your full peer review and any attached files.

Reviewer #1: No

Reviewer #2: No

---

## [Author Response · Author response to Decision Letter 0]

27 Nov 2019

Revision of PONE-D-19-23445 manuscript entitled “Changing perioperative prophylaxis during antibiotic therapy and iterative debridement for orthopedic infections?” by Wuarin et al. 

Dear Dr Daniel Pérez-Prieto

We thank you for your email of 1 November 2019 in which you invite us to submit a revised version of our above-mentioned manuscript. 

We also want to thank both Reviewers for the careful reading and suggestions provided. Of note, we incorporated all their suggestions into the manuscript. 

All changes are tracked. All authors agreed to the final version of the paper.

Thanking you in advance for your consideration of our revised manuscript and we look forward to the final decision of the Editorial Committee.

Yours most sincerely,

Ilker Uçkay, MD 

Editors Comments

A marked-up copy of your manuscript that highlights changes made to the original version. This file should be uploaded as separate file and labeled 'Revised Manuscript with Track Changes'.

Answer: Done.

An unmarked version of your revised paper without tracked changes. This file should be uploaded as separate file and labeled 'Manuscript'.

Answer: Done.

Answer: We now adapted the manuscript according to this editorial guidance.

2. Please include your tables as part of your main manuscript and remove the individual files.

Answer: Done as advised. However, we have five large Tables. By fitting them into the manuscript, the Tables become difficult to read. Therefore, we take the liberty to resumes all five Tables on a separate file, too. 

Answer: We have no Supporting Information.

Reviewers' comments:

1. Is the manuscript technically sound, and do the data support the conclusions?

Reviewer #1: Yes

Reviewer #2: Yes

Answer: We thank both Reviewers.

2. Has the statistical analysis been performed appropriately and rigorously? 

Reviewer #1: Yes

Reviewer #2: Yes

Answer: We thank both Reviewers.

3. Have the authors made all data underlying the findings in their manuscript fully available?

Reviewer #1: Yes

Reviewer #2: Yes

Answer: We thank both Reviewers.

4. Is the manuscript presented in an intelligible fashion and written in standard English?

Reviewer #1: Yes

Reviewer #2: Yes

Answer: We thank both Reviewers.

Reviewer #1: 

The article titled: “Changing perioperative prophylaxis during antibiotic therapy and iterative debridement for orthopedic infections?” Is a description of the microbiological findings in patients with orthopedic infections that required a new re-operation for failure of the infection during antibiotic treatment. 

Answer: Yes. Exactly. Thank you.

This is a very original work, well written and with a large number of patients included in the analysis from a retrospective database. 

Answer: Thank you very much. We introduce this sentence now on page 16, lines 272-273.

I have some comments:

1.- In the abstract and in your conclusions, you highlight that (abstract) “Selection of new pathogens resistant to ongoing antibiotic therapy occurs infrequently” (conclusion) “According to our cohort of 2480 adult patients with orthopedic infections, new SSIs occur at ten percent’s risk during iterative debridement and concomitant antibiotic therapy.” From a clinical perspective, I think that the impressive data is that from 862 patients that required a re-debridement 507 (59%) had a positive culture and in 265 (52%) the isolated microorganism is different from the first one. This means that from all that required a re-debridement 862, 30.7% (265) had a different pathogen and indeed this is not a minor problem particularly considered that the microorganisms were often more resistant.

Answer: Exactly. We like this (alternative) way of resuming by Reviewer 1. We now introduce his/her phrasing into the Discussion (page 16, lines 276-280).

2.- According to material and methods section: “We excluded cases that were amputated [11], cases with antibiotic-free windows before re- debridement…” so all patients included in the study were under antibiotic treatment. 

Answer: Yes; besides the very first debridement for the orthopedic infection (when the antibiotics started after the intraoperative microbiological samplings). We explicitly say this in the Methods (page 5, lines 98-100) and repeat it at various parts of the manuscript.

Then I don't understand what you say in results section: “Table 2 shows clinical variables related to new SSIs. Current antibiotic administration and…” All patients were under antibiotic treatment? 

Answer: Yes. The entire analyzed population was already under systemic antibiotic therapy (page 5, lines 98-100). 

In Table 2 you show different antibiotics (penicillin, cephalosporins, …) but no one was significantly associated with new G+ or G-.

Answer: Indeed. No antibiotic class (in stratified analyses) was associated with the Gram-staining of the new pathogens. The analyzed population were entirely under some systemic antibiotic therapy (page 5, lines 98-100). 

3.- In the Methods section for the multivariate analysis you mention: “An unmatched multivariate logistic regression analysis determined associations with the outcome “SSI resistant to antibiotic therapy”.” So, until now the endpoint of the study (Table 2) was new SSI and for the multivariate adjustment you change to antibiotic-resistant (to the current treatment) SSI. 

Answer: No. In the multivariate analyses, the outcome variables were always “resistant microorganisms” (as explained in “Statistical analyses”). We now reword better on page 10, line 193 and on page 14, lines 246-253.

It is reasonable to expect that the majority of new pathogens were resistant to current antibiotics but this was indeed the case? Could you confirm that all new pathogens were resistant to current antibiotic treatment. 

Answer: No, not every new pathogen was resistant to current (prior) antibiotics. We already say it on page 8, line 180: "resistant to current antibiotics in 174 cases (174/507; 34%)". We also add now the number of new susceptible pathogens and display some key, non-resistant, pathogens in Figure 1, too. 

In addition, in multivariate adjustment you mention: “We confirmed that current antibiotic use was associated with new antibiotic-resistant SSIs (odds ratio 1.6, 95% confidence interval 1.2-2.2), …” but in Table 5 there is no reference to current antibiotic variable? Indeed, all the patients were under antibiotic treatment?

Answer: Yes. Every patient in the corresponding analysis was under systemic antibiotic therapy. Therefore, there is no sense to introduce a variable “antibiotic” into the final model, if every patient already is under antibiotics. We now explain this aspect better in Results (page 14, lines 250-253), and repeat it as Limitation (page 19, lines 335-338).

We equally modify the former sentence "We confirmed that current antibiotic use was associated with"… to a new phrasing (page 14, lines 247-248).

4.- The reason for a new infection (isolation of a different pathogen from the first one) could be the consequence of 1.- miss identification during the first surgery for infection (unlikely considering that nowadays, and for sure in your expert hospital, several samples are collected during surgery, 2.- new contamination during previous surgery for infection (first or second or third… surgery) and 3.- superinfection of the wound after surgery. Considering only 2 and 3 as possible, the third is not modifiable by antibiotic prophylaxis but the number 2 could be significantly reduced. 

Answer: Thank you very much. This is a probably a better résumé than we initially wrote. We thank Reviewer 1 for this short résumé and introduce it into the Discussion (page 17, lines 297-302).

In the text you talk about prophylaxis for the new debridement but this is not correct. New pathogens are indeed new infections so you have to talk about additional empirical treatment when a patient requires a new debridement.

Answer: We beg to differ. The Reviewer 1 may think that we mention replacing a current antibiotic treatment by a new one; basing on presumed changes in microbiology and continuing with this new regimen during several days in a therapeutic manner. 

No. Wherever we talk about prophylaxis, we truly mean additional prophylaxis under current therapeutic antibiotic therapy. Just one dose of a perioperative antibiotic prophylaxis, which prevents new infections, and is not continued after surgery. We now make this distinction clearer in the Methods (page 5, lines 101-105).

A different point is whether the previous debridement to the current one with no pathogens, the same pathogen or a new pathogen, required a different prophylaxis to avoid contamination by a different pathogen (if the reason for a new infection is a contamination during the previous debridement, option number 2). 

Answer: In our study population, we had 83 different antibiotic regimens prior to the occurrence of 273 new pathogens (page 6, lines 133-134; Table 4). A given episode could witness different antibiotic regimens throughout the therapeutic course. To these microbiological differences, we also add a variety of 867 new antibiotic regimens (page 7, line 145). Such resulting two-by-two tables become too much detailed and individualized to be displayed in this manuscript. 

We have to group the different subpopulations, which we did. However, Reviewer 1 is right in a conceptual way. We acknowledge it now in the "Limitation" section (Forth limitation; page 19, lines 358-362).

Could you evaluate the antibiotics received during the prior debridement to the one with or without a new pathogen? This would be really interesting.

Answer: The same question and answer as above. Of note, all patients during iterative surgeries were already under systemic antibiotic therapy (page 5, lines 98-100). Therefore, we cannot analyze if a prior antibiotic therapy per se was associated with an enhanced risk of new infections (page 19, lines 355-358). 

We can only analyze of a particular antibiotic class was associated with a particular group of new pathogens, which we already displayed on Table 2 and Figure 1.

Reviewer #2: 

I read with interest the paper. It is a comprehensive analysis of complex set of data of a large cohort of patients with orthopedic infection with a clear aim. It needs a huge effort to organize this type of data. And it is nearly impossible to study the hypothesis in any other prospective and more controlled way. The aim of the study is clearly defined and important. The issue of antibiotic prophylaxis and treatment in unsuccessful orthopedic infections is a very important and still unsolved topic touching every specialist involved in treatment of this complex pathology.

Answer: Thank you very much. We now use the last sentence in our revised manuscript (page 16, lines 283-284).

Despite the complexity of the cohort data the authors were able to organize it in a systematic way. The presenting cohort is thus clearly outlined in the text and trough the tables. Despite the study did not give a clear answer to the original question it still revealed the complexity of the clinical field and gave thought lines for the readers to integrate in their clinical decisions.

Answer: We thank you very much.

Page 3 last paragraph: Why were there revisions in satisfactory clinical evolution?

Answer: Planned second or third looks in order to decrease the bacterial inoculum, for which the indication has been decided already during the first debridement. Unfortunately, it is very common in surgical settings to re-operate the patient in a planned way in order to (wrongly) accelerate the healing, especially when the patient was instable or has other parameters to consider. We now underline this possibility of “planned reinterventions” on page 3, lines 64-65.

Page 4 last paragraph: Why episodes with new organisms that did not change the antibiotics were interpreted as contamination, just because they were sensitive to original antibiotic therapy? 

Answer: There are three answers, which we all explain better in this revised version of our paper:

a) The interpretation of a “contamination” clinically based on a microbiological finding during a good evolution, meaning that the newly detected bacteria had no new clinical (worse) impact (page 4, lines 83-87).

b) In contrast, pathogens sensitive to original antibiotic therapy can truly cause a clinical worsening, but are identified as a new infectious constellation (page 4, lines 87-88).

c) Theoretically, new bacteria can also be a new infection that was not severe enough to worsen the clinical evolution (yet). In that sense, when there is good clinical evolution, it is impossible to distinguish between colonization and clinical new infection, which we now acknowledge it the “Limitation section” (page 19, lines 349-352).

According to the definition of the infection it should thus be mentioned that only high-grade infections were included.

Answer: Yes. “Clinically moderate and severe” infections. We now add this prerequisite to the inclusion criteria (page 4, line 79).

Page 10 second paragraph: bundle of measures [1-3]. "Thus, avoiding unnecessary re-debridement, evidence-based surgical indications and techniques, and experienced surgical skills are certainly as important as mere addition of a new molecule." This sentence is very brave and to me lacking evidence besides common thinking.

Answer: Yes, this is common thinking. And we have now deleted this sentence entirely.

Page 10 second paragraph: "At the same time, numerous reported transient kidney injuries by aminoglycosides [27] or combined vancomycin prophylaxis [28] in orthopedic surgery." The sentence has no verb.

Answer: Yes, thank you. The missing verb was “documented” (page 18, line 332).

6. PLOS authors have the option to publish the peer review history of their article (what does this mean?). If published, this will include your full peer review and any attached files. Answer: It is up to you. I do not mind to have my personal identity to be public.

Do you want your identity to be public for this peer review? For information about this choice, including consent withdrawal, please see our Privacy Policy.

Reviewer #1: No

Reviewer #2: No

Answer: Yes, we have uploaded them.

---

## [Editor Report · Decision Letter 1]

5 Dec 2019

Changing perioperative prophylaxis during antibiotic therapy and iterative debridement for orthopedic infections?

PONE-D-19-23445R1

Dear Dr. Uçkay,

We are pleased to inform you that your manuscript has been judged scientifically suitable for publication and will be formally accepted for publication once it complies with all outstanding technical requirements.

With kind regards,

Daniel Pérez-Prieto, PhD

Academic Editor

PLOS ONE
---

## [Editor Report · Acceptance letter]

10 Dec 2019

PONE-D-19-23445R1 

Changing perioperative prophylaxis during antibiotic therapy and iterative debridement for orthopedic infections? 

Dear Dr. Uçkay:

I am pleased to inform you that your manuscript has been deemed suitable for publication in PLOS ONE. Congratulations! Your manuscript is now with our production department. 

With kind regards,

on behalf of

Dr. Daniel Pérez-Prieto 

Academic Editor

PLOS ONE